# Abbreviated MRI for Comprehensive Regional Lymph Node Staging during Pre-Operative Breast MRI

**DOI:** 10.3390/cancers15061859

**Published:** 2023-03-20

**Authors:** Maike Bode, Simone Schrading, Arghavan Masoumi, Stephanie Morscheid, Sabine Schacht, Timm Dirrichs, Nadine Gaisa, Elmar Stickeler, Christiane K. Kuhl

**Affiliations:** 1Department of Diagnostic and Interventional Radiology, University Hospital Aachen, 52074 Aachen, Germany; 2Department of Pathology, University Hospital Aachen, 52074 Aachen, Germany; 3Department of Gynecology and Obstetrics, University Hospital Aachen, 52074 Aachen, Germany

**Keywords:** breast MRI, breast cancer, nodal disease, lymph node metastases

## Abstract

**Simple Summary:**

The clinically relevant information that guides further surgical management of the axilla in women with breast cancer is the distinction between non-significant (N0–N1) versus significant (≥N2) lymph node metastases. In women with newly diagnosed breast cancer, MRI is increasingly used to determine the local extent of disease in the breast. The aim of our retrospective study on 414 patients who underwent routine breast MRI for local staging of the breast was to determine whether an abbreviated protocol for regional lymph node staging is sufficient to identify patients with significant nodal disease. Our results demonstrated that a single 3 min coronal T1-weighted sequence, acquired with the system’s built-in body coil, covering the chest wall, axilla, and supra- and infraclavicular region, helped rule out the presence of significant nodal disease with a NPV of 98.8% [97.0–100%]. False-positive findings were mostly caused by patients with positive, but non-significant, lymph node metastases (N1).

**Abstract:**

Background: The detection of regional lymph node metastases (LNM), in particular significant LNM (≥N2), is important to guide treatment decisions in women with breast cancer. The purpose of this study was to determine whether a coronal pulse sequence as part of pre-operative breast MRI is useful to identify women without significant LNM. Material: Retrospective study between January 2017 and December 2019 on 414 consecutive women with breast cancer who underwent pre-operative breast MRI on a 1.5 T system. For lymph node (LN) staging, a coronal pre-contrast non-fat-suppressed T1-weighted TSE sequence was acquired with the system’s built-in body coil, covering the chest wall; acquisition time 3:12 min. Two radiologists rated the likelihood of LNM on a 3-point scale (absent/possible/present). Validation was obtained by histology from sentinel LN biopsy, axillary LN dissection, and/or PET/CT. Results: 368/414 women were staged to have no or non-significant LNM (pN0 in 282/414, pN1 in 86/414), and significant LNM (≥pN2) in 46/414. For identification of women with significant LNM, MRI was true-positive in 42/46, false-negative in 4/46, true-negative in 327/368, and false-positive in 41/83, the latter mostly caused by women with N1-disease (38/41), yielding an NPV and PPV for significant LNM of 98.8% [95%-CI: 97.0–100%] and 50.6% [43.1–58.1%], respectively. Conclusions: A 3 min coronal T1-weighted pulse sequence covering the chest wall as part of pre-operative breast MRI is useful to rule out significant LNM with high NPV. Where MRI staging is positive for significant LNM, additional work-up is indicated to improve the distinction of N1 and N2 disease.

## 1. Introduction

Tumor biology such as estrogen and progesterone receptor status, Her2-receptor status and Ki-67 proliferation rates are used to determine the need for systemic treatment of patients with newly diagnosed invasive breast cancer. However, information on regional lymph nodes (LN) integrity is still required and usually collected through sentinel lymph node biopsy (SLNB) [1]. Until recently, the results of the SLNB were used to decide whether or not axillary surgery is needed, with patients with positive SLNB proceeding to axillary clearance [2]. However, axillary surgery is associated with significant morbidity including lymphatic edema, paraesthesia, and possibly with permanent impairment of upper limb motility [3,4,5]. Therefore, in 2005, the American College of Surgeons Oncology Group (ACOSOG) published the results of the ACOSOG Z0011 study, on women with clinical T1-T2 invasive breast cancer with SLNB-based lymph node metastases (LNM) categorized as ≤N1, and without palpable nodal adenopathy, randomized to either receive a complete axillary lymph node dissection (ALND), or to no further surgical treatment of the axilla. The results of the study indicated a similar 5-year as well as 10-year disease-free and overall survival for both groups, indicating that ALND after a positive SLNB with such “non-significant” LN disease would not be needed [6,7,8,9].

Thus, with Z0011, the clinically relevant information that guides further surgical management of the axilla is not any more the distinction between negative (N0) versus positive (≥N1) LN–but between no and non-significant (N0–N1) versus significant (≥N2) LNM. Therefore, in the era of SLNB and after the Z0011 study, the task of imaging for regional LN staging in women with newly diagnosed invasive breast cancer changed fundamentally. To guide further systemic treatment, the presence of any LNM is relevant; this information is established via SLNB, which allows the identification even of isolated tumor cells classified as N0, or deposits as small as 2 mm that are sufficient to stage a lymph node as N1. Imaging has no or little role in this regard. However, to guide further surgical treatment of the axilla, with the new Z0011 paradigm, it is only important to identify patients with nodal disease beyond N1. This is information structural imaging can be expected to provide. Moreover, since the Z0011 trial cohort included only a relatively small number of patients with advanced nodal disease, and since Z0011 was the only randomized trial on this issue so far, there are still concerns that women who are treated according to the ACOSOG Z0011 pathway may have undetected significant LNM which could give rise to local morbidity and mortality [3].

Prior approaches to integrate axillary LN staging in routine pre-operative breast MRI used the breast coil for this purpose [10]. One challenge of this approach, however, is the limited axillary coverage of even contemporary breast coils [11,12]. Long examination times of high-resolution axillary imaging add to lengthy acquisition protocols and, thus, to overall costs. Finally, regional LN include not only the axillary, but also the parasternal and infra- and supraclavicular basin [13].

Accordingly, our approach was to not use the breast coil, but the system’s built-in body coil for imaging of the chest wall including both sides of the parasternal, the axillary and infra- and supraclavicular regions for comprehensive regional LN staging.

The aim of this study was to investigate whether such a short pulse sequence is useful to help identify patients without significant LNM who can proceed according to the ACOSOG Z0011 pathway.

## 2. Materials and Methods

### 2.1. Study Design and Inclusion and Exclusion Criteria

This retrospective study was conducted in accordance with the Declaration of Helsinki between January 2017 and December 2019 and was approved by the Institutional Review Board of RWTH University Aachen (EK205/17); the need for written consent was waived. Consecutive women diagnosed with invasive breast cancer who underwent pre-operative breast MRI were included. Exclusion criteria were women who had already undergone neoadjuvant systemic treatment, or women with pure ductal carcinoma in situ. Women were also excluded if the reference standard (histopathology for axillary LN, PET/CT for internal mammary and supraclavicular LN) was not available for nodal staging. Of note, this study was conducted before the SARS-COV-2 pandemic and population vaccination campaigns.

### 2.2. MR Imaging Protocol

Bilateral breast MRI was performed in prone position and according to standardized protocol [14] on a 1.5 T system (Achieva, Philips Medical System) with a four-channel bilateral breast coil (Invivo, Gainesville, FL, USA). The abbreviated breast protocol consisted of an axial T2-weighted turbo spin echo (TSE) sequence and an axial dynamic contrast-enhanced series acquired before and three times after contrast agent injection.

For regional LN staging, just before the dynamic series, a coronal T1-weighted TSE sequence (repetition time/echo time [TR/TE]: 550/15), with a field of view of 370 mm, a non-interpolated 320 × 300 acquisition matrix, a slice thickness of 4 mm and an acquisition time of 3:12 s was acquired with the system’s built-in body coil. This is the coil that is usually used to transmit radiofrequency pulses, with the respective local surface coils serving as receivers–which is, e.g., the breast coil for breast MRI. For the coronal T1-weighted pulse sequence, however, this body coil was also used as a receiver coil. This sequence was prescribed to cover the anatomic areas behind the breast, i.e., the chest wall including the parasternal, infra- and supraclavicular region, and axilla.

### 2.3. MRI Evaluation of Lymph Node Status

Image sets were retrospectively read in consensus by two breast radiologists with 11 and 6 years of experience. Readers were aware of the histopathological diagnosis of invasive breast cancer. For study purpose, the likelihood of suspicious LN in the axillary, parasternal, and infra- and supraclavicular regions were assessed on a 3-point scale ranging from category 1 = definitely absent to category 3 = definitely positive.

Criteria to categorize LN as metastatic were their size measured in short axis, their morphology including their shape, presence of a fatty hilus, and their signal intensity on T1-weighted imaging (all LN basins) and T2-weighted imaging (axillary LN). For assessment of axillary LN in women with unilateral breast cancer, their signal intensity was visually compared to the respective contralateral side.

In detail, presence of LNM was categorized as “definitely absent” (category 1) for axillary LN if their size was ≤1 cm in short axis, they showed a fatty hilum, were oval or kidney shaped, and had an internal signal intensity similar to the contralateral LN (Figure 1A). For parasternal and infra- and supraclavicular LN, this was the case when there were no LN visible at all, or a single LN smaller than 5 mm in short axis, intermediately low signal intensity, and visible hilum (Figure 1B–F).

The presence of LNM was categorized as “possibly present” (category 2) for axillary LN if their size was ≤1 cm in short axis but they showed a suspicious morphology or signal intensity: Loss of fatty hilum, and/or roundish shape, and/or hypointense signal intensity compared to the contralateral LN. For parasternal or infra/supraclavicular LN, this was the case when in this region, a solitary LN was visible with a short axis larger than 5 mm and/or no fatty hilum.

The presence of LNM was categorized as ”definitely positive“ (category 3), when axillary LN had a short-axis size > 1 cm and they showed at least one of the suspicious morphology features described above (Figure 2A,C,D). For parasternal or infra/supraclavicular LN, this was the case when there were more than one LN larger than 5 mm in short axis and/or the LN were markedly hypointense (Figure 2B–F).

### 2.4. Validation of Lymph Node Diagnoses

The work-up of axillary, infra/supraclavicular or parasternal LNM followed a standardized protocol. Axillary LN underwent staging by SLNB and (if beyond ACOSOG Z0011 criteria) ALND. A composite reference standard was used to validate imaging findings of the internal mammary and supraclavicular nodes; this was discussed in the respective patients’ multi-disciplinary tumor board and was based on all information available on the given patient, usually including diagnostic information from PET/CT and/or US-guided biopsy.

The nodal stage was categorized according to the 8th edition of TNM classification of malignant tumors [15]. pN0, negative; pN1, metastases in 1–3 axillary and/or internal mammary LN; pN2, metastases in 4–9 axillary LN, and tumor cell deposits are larger than 2 mm in at least one LN, or internal mammary LN; pN3, metastases in ≥10 axillary LN, internal mammary LN or any LN in the infra- or supraclavicular region.

### 2.5. Data Analysis

Statistical analysis was performed using IBM SPSS Statistics (version 28.0, Amronk, NY, USA). For qualitative findings, frequencies and their distributions were determined. Quantitative data are presented as means with standard deviations and ranges.

A diagnosis of “possibly” or “definitely” positive LN on MR imaging was considered as “test-positive”.

In accordance with the ACOSOG Z0011 pathway, the nodal status was dichotomized into “clinically significant nodal disease”, defined as nodal disease beyond stage pN1, and “clinically non-significant” nodal disease, defined as no nodal disease (pN0) or limited nodal disease (pN1). The diagnostic performance was assessed by calculating the sensitivity, specificity, NPV, and PPV. A total of 95% CIs were reported for the diagnostic parameters. *p*-values and CIs were two-sided and 0.05 was used as a cut-off for significance.

## 3. Results

During the study period, a total of 10,164 women underwent breast MRI in our facility. A coronal T1-weighted TSE sequence as part of the pre-operative MRI was performed on 699 women with pre-invasive/invasive breast cancer. Of these, 136 women had pure ductal carcinoma in situ and 25 women underwent MRI after neoadjuvant systemic treatment leading to study exclusion. In the remaining 538 women with invasive breast cancer, histopathologic details on SLNB and/or ALND for axillary nodal staging or PET/CT for parasternal/supraclavicular LN validation were available in 414 women (Figure 3).

The mean age of patients was 59.51 ± 11.06 years (range, 26 to 82 years). Patient and tumor characteristics are summarized in Table 1.

Of the 414 patients, 282 (68.1%) were finally staged to be without LNM (pN0); 86 (20.8%) were staged to have pN1; thus, a total 368/414 (88.9%) were staged to have no or insignificant LNM (pN0 or pN1). The remaining 46/414 patients (11.1%) were finally categorized as having significant LNM (stage pN2 or higher) (Table 2).

MRI was positive for LNM in a total 83/414 patients (20.0%), of whom 3/83 were finally staged as LN negative (pN0), 38/83 were staged as pN1, and 42/83 were staged as >pN1. Thus, MRI was true-positive for diagnosis of any LNM in 80/83 patients, and true-positive for diagnosis of significant LNM in 42/83 (Figure 2). MRI was false-positive regarding the identification of any stage of LNM in 3/83 patients, and was false-positive regarding the identification of patients with significant LNM in 41/83 patients, including 38 patients who were pN1. Accordingly, regarding the identification of patients with significant LNM, the vast majority of false-positive MRI-diagnoses were caused by patients with non-significant (N1) disease.

MRI was negative for LNM in a total 331/414 patients (80.0%), of whom 279/331 were finally staged as LN negative (pN0) (Figure 1), 48/331 were finally staged as having non-significant LNM (pN1), and 4/331 to have significant LNM (>pN1). Thus, MRI was true-negative in patients without any LNM (pN0) in 279/282, and true-negative for patients without significant LNM (pN0 + pN1) in 327/368 patients. MRI was false-negative for any stage of LNM in 52/331 patients, and false-negative for significant LNM in 4/331 patients.

Accordingly, MRI offered an overall sensitivity, specificity, NPV and PPV for any stage LNM of 60.6% (95% CI: 51.7–69.0; 80/132), 98.9% (95% CI: 96.9–99.8; 279/282), 84.3% (95% CI: 81.3–86.9; 279/331), and 96.4% (95% CI: 89.6–98.8; 80/83).

Regarding the identification of patients with significant nodal disease, MRI offered a sensitivity, specificity, NPV, and PPV of 91.3% (95% CI: 79.2–97.6; 42/46), 88.9% (95% CI: 85.2–91.9; 327/368), 98.8% (95% CI: 97.0–99.5; 327/331), and 50.6% (95% CI: 43.1–58.1; 42/83) (Table 3).

## 4. Discussion

In this study, we found that a short additional coronal T1-weighted pulse sequence, acquired with the system’s built-in body coil, can be integrated in a routine pre-operative breast MRI protocol, and can help exclude clinically significant (≥pN2) LNM with high negative predictive value. This suggests that standard pre-operative breast MRI can be used not only to map the local extent of disease within the breast, but also to provide accurate information on regional LN staging that is important to guide patient management.

Until the introduction of SLNB, all women with newly diagnosed invasive breast cancer underwent full ALND for both diagnostic and therapeutic purposes [16,17]. To avoid the substantial morbidity associated with ALND, the concept of SLNB was introduced [18,19]. Until the results of the ACOSOG Z0011-study, all women with positive SLNB still had to undergo full ALND. Since then, women with invasive breast cancers no larger than 5 cm in diameter, without clinically palpable axillary or parasternal nodes, and with 1–2 positive nodes on SLNB, are able to forgo axillary dissection [20,21].

Information on nodal disease is still a decisive factor to guide systemic treatment in particular in patients with ER-positive, HER-2 negative disease [22]. Since systemic treatment considers all stages of LNM including less advanced manifestations of N1 disease–defined as presence of metastatic deposits of at small as 2 mm diameter in at least one and up to three axillary or parasternal nodes—this information is usually obtained by biopsy. Since SLNB is well-tolerated and enables not only a reliable diagnosis also of small cancer cell deposits, but also subtyping of LNM, it is unlikely that imaging will soon replace this.

However, the decision whether or not to recommend axillary surgery is currently based on the distinction of N0/N1 vs. N2 disease, the latter defined as LNM in at least four or more axillary or parasternal LN. Thus, there is a clinical need for imaging methods that can help establish this differential diagnosis. Our results suggest that a single additional pulse sequence, added to an breast MRI local staging protocol, can provide this information and help to rule out presence of significant nodal disease even in the parasternal or infra- and supraclavicular region with high negative predictive value of 98.8%.

The usual candidate for detection of LNM is ultrasound, which also offers the possibility of immediate US-guided biopsy. However, ultrasound is operator-dependent—with published levels of sensitivity and specificity ranging from 26% to 76%, and from 88% to 98%, respectively [23,24]. Ultrasound is less accurate in identifying parasternal LN and is not commonly used to depict infra- or supraclavicular nodes (18). Our results demonstrated that accurate information on regional axillary staging, in particular the exclusion of significant nodal disease, can be an integral part of routine breast MRI staging protocols, obviating the need for additional ultrasound in many patients. Compared with ultrasound, abbreviated LN staging as part of pre-operative breast MRI allows accurate staging of all regional LN basins including not only the axilla but also the parasternal and supra- and infraclavicular basins. Our data suggest that ultrasound could be reserved for patients with positive MRI findings.

For regional LN staging, we used a non-fat-suppressed T1-weighted coronal acquisition because this enables the delineation and morphologic characterization of axillary LN (level I-III), parasternal LN, as well as infra- and supraclavicular LN with high contrast against the fatty tissue of the parasternal region, the armpit, and the infra- and supraclavicular region. Since contrast enhancement was observed in all LN regardless of the presence or absence of metastases, and since enhancing LN may become isointense to surrounding fatty tissue, we acquired the pulse sequence for LN staging before contrast agent injection [25]. As an alternative, one could suppress the signal from fat to highlight enhancing LN on post-contrast images, but this takes extra time, is associated with additional cardiac pulsation artefacts, and offers no benefit compared with pre-contrast imaging [26]. The coronal plane is known to be useful to depict parasternal and infra- and supraclavicular nodes (Figure 1E,F), and requires much fewer sections to cover the anatomic region than axial images.

Our results suggest that such a short pulse sequence offers a similar diagnostic accuracy for axillary LN staging as do those using the breast coil. Van Nijnatten et al. reported an NPV for advanced axillary disease of 99.1–99.3% [27] for axillary LN. Again, however, our approach allows the detection of LN not only in the axilla, but also in the parasternal and supra- and infraclavicular region, for a far more comprehensive assessment of regional nodal disease.

Even today, adjuvant radiotherapy of the supra- and infraclavicular and—in women with medial tumor location—the parasternal basins is routinely used in women who do not undergo axillary dissection. With the abbreviated LN staging protocol, we are able to accurately detect also parasternal and supra- and infraclavicular nodes. We believe that future clinical trials should employ such simple MR staging methods to select a group of women who can safely forgo such radiotherapy.

About half of the women called positive for LNM on MRI did have positive nodes, but only non-significant stages (N1). This is likely due to the fact that readers were not asked to distinguish between significant and non-significant disease, but were asked to report any positive or possibly positive LN, regardless of the LN tumor burden. As a clinical consequence, however, this means that once abbreviated MRI staging is positive for LNs, additional imaging is needed to distinguish between non-significant and significant LN disease.

Fifty-two women were categorized as “negative” on abbreviated MRI, but had positive LN on SLNB; in most of these cases (48/52), patients had N1-disease. This was not unexpected and simply underscores the fact that imaging, including our approach, is not useful to safely rule out stage N1. However, in 4/52 patients, MRI was (false-)negative, although substantial (stage N2) LN metastases were present. These patients had disseminated metastases in up to 10 LNs without LN enlargement or suspicious lymph node morphology. Of note, the TNM classification considers tumor cell deposits larger than 2 mm as positive.

LNM in women with breast cancer, if undetected, can cause significant local morbidity, and have important prognostic implications. Women who undergo treatment according to the ACOSOG Z-0011 pathway will not undergo axillary surgery, which can cause LNM outside the sentinel node to remain undetected. Abbreviated regional LN staging integrated in routine pre-operative breast MRI could help identify these women and, thus, help avoid such morbidity and mortality. Further research should address the clinical integration of abbreviated regional LN staging.

Our study had several limitations. First, we were not able to perform a lesion-by-lesion correlation of imaging findings and findings of SLNB or ALND. This limitation, however, is true for most published studies on nodal staging. Second, we did not have histological verification for all patients suspected to have parasternal or supra- or infaclavicular disease, but had to rely on PET/CT imaging for correlation. Third, we included consecutive patients with invasive breast cancers, some of which were finally staged beyond the ACOSOG Z-0011 group of T1-T2 cancers. Finally, and of major importance, our study was conducted just prior to the COVID-19 pandemic. With the currently ongoing vaccination campaign, the majority of women who present for screening or staging will have hyperplastic LN [28]. Thus, our results may not be fully translated to current clinical practice until the pandemic situation is overcome.

## 5. Conclusions

We found that an abbreviated lymph node staging protocol, acquired with the system’s build-in-coil, can be integrated in routine pre-operative breast MRI, and is useful to exclude significant nodal disease with high negative predictive value.

## Figures and Tables

**Figure 1 cancers-15-01859-f001:**
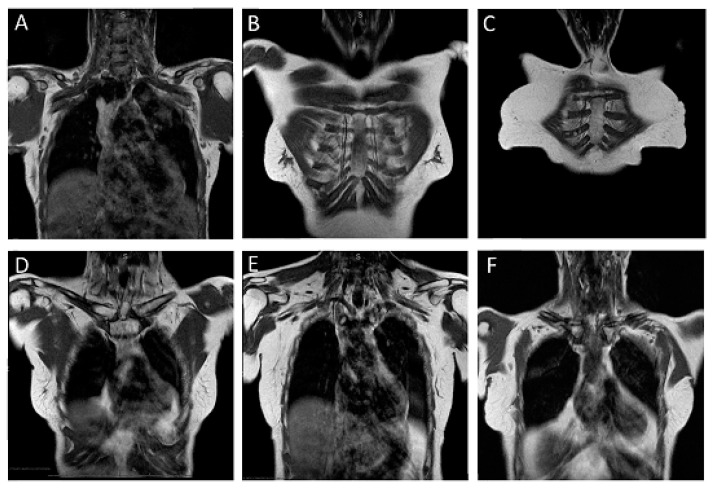
Normal lymph node basins. (**A**): normal axillary basins in a 70-year-old woman with newly diagnosed luminal A breast cancer of her left breast. (**B**): normal parasternal region in a 67-year-old patient with newly diagnosed Her2-positive luminal B breast cancer of the left breast. (**C**): normal parasternal lymph node in a 66-year-old woman with newly diagnosed luminal A breast cancer of her left breast. (**D**,**E**): normal infra- and supraclavicular basin in a 57-year-old woman with newly diagnosed luminal A breast cancer of her right breast. (**F**): normal lymph nodes in the infra-clavicular basin on the left in a 50-year-old woman with newly diagnosed Her2-negative luminal B breast cancer of her left breast.

**Figure 2 cancers-15-01859-f002:**
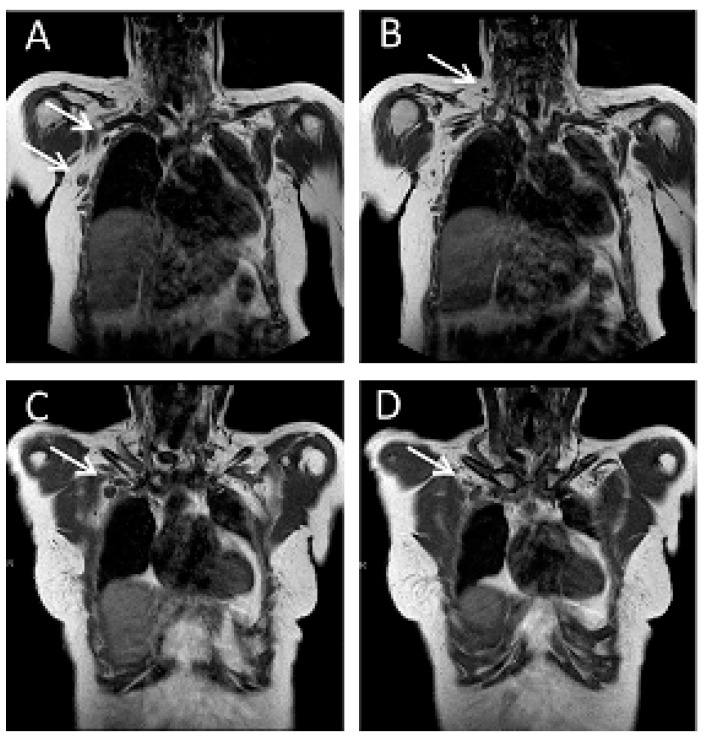
Pathologic nodal findings. (**A**,**B**): 52-year-old women with newly diagnosed Her2-negative luminal B breast cancer of the right breast. A, B, two consecutive sections of the coronal T1-weighted TSE sequence. Coronal T1-weighted TSE images depict axillary lymph node metastases level 1 and level 3 (**A**), and a supra-clavicular lymph node (**B**). (**C**,**D**): 32-year-old female patient with newly diagnosed triple-negative breast cancer of the right breast. Coronal T1-weighted TSE image depicts infraclavicular lymph nodes. (**E**,**F**): Parasternal lymph node metastasis in a 44-year old female women with newly diagnosed triple negative breast cancer of the right breast. (**E**), coronal T1-weighted TSE image obtained with the body coil, (**F**), axial pre-contrast T1-weighted image obtained with the breast coil. Note that the lymph node is far better visible on the coronal image compared to the axial image. Arrows indicate the lymph nodes.

**Figure 3 cancers-15-01859-f003:**
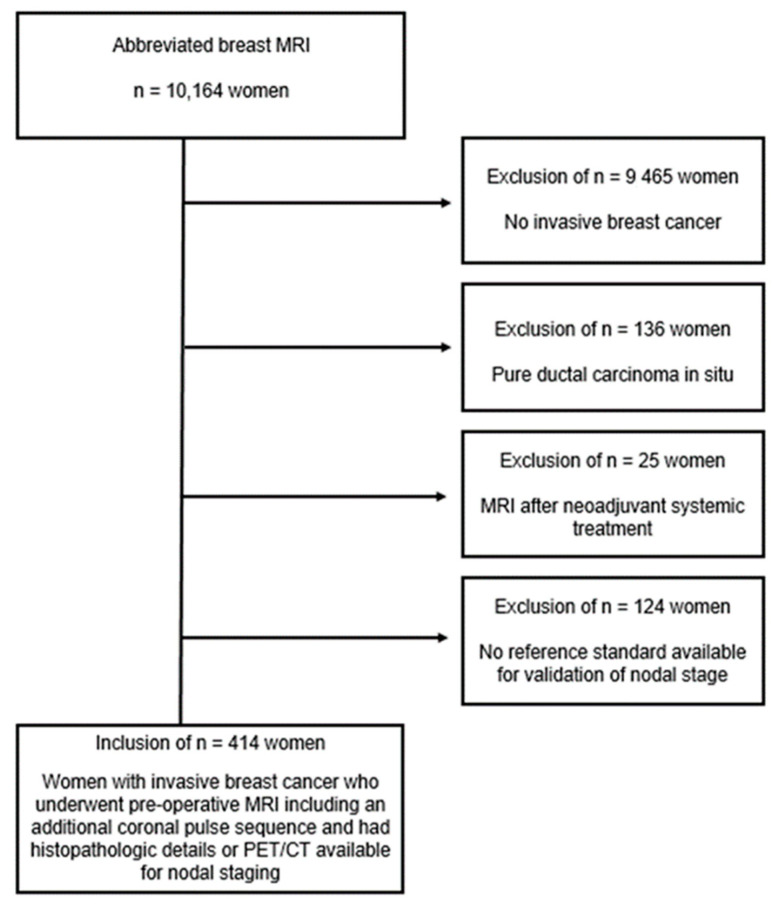
Flow diagram of the study inclusion procedure.

**Table 1 cancers-15-01859-t001:** Patient demographics and characteristics of invasive breast cancer. Note—data in brackets are percentages. LNM = lymph node metastases, n/a = not available, NST = No special type, SD = standard deviation.

Characteristic		Women with No or Non-Significant LNM (pN0 or pN1) (n = 368)	Women with Significant LNM (≥pN2) (n = 46)	All Women (n = 414)
Demographic features				
Age	mean ± SD, years	59.5 ± 11.1	59.6 ± 11.1	59.5 ± 11.1
Menopausal status				
	Premenopausal	92 (25.0)	16 (3.5)	108 (26.1)
	Postmenopausal	276 (75.0)	30 (6.5)	306 (73.9)
Familial risk				
	Average	236 (64.1)	28 (60.9)	264 (63.8)
	Moderate	72 (19.6)	8 (17.4)	80 (19.3)
	High	45 (12.2)	5 (10.9)	50 (12.1)
	n/a	15 (4.1)	5 (10.9)	20 (4.8)
Characteristics of the newly diagnosed breast cancer				
Histologic subtype				
	NST	246 (66.8)	34 (73.9)	280 (67.6)
	Invasive lobular	93 (25.3)	12 (26.1)	105 (25.4)
	Other	29 (8.0)	-	29 (7)
Molecular subtype				
	Luminal A	172 (46.7)	15 (32.6)	187 (45.2)
	Luminal B	96 (26.0)	12 (26.1)	108 (26.1)
	Her2 type	79 (21.5)	16 (34.8)	95 (22.9)
	Triple negative	20 (5.4)	4 (8.7)	24 (5.8)
	n/a	3 (0.8)	-	3 (0.7)
Nuclear grade				
	Low	45 (12.2)	1 (2.2)	46 (11.1)
	Intermediate	245 (66.6)	27 (58.7)	272 (65.7)
	High	78 (21.2)	18 (39.1)	96 (23.2)
Stage information				
T-stage				
	T1	228 (61.2)	14 (30.4)	244 (58.9)
	T2	111 (30.2)	15 (32.6)	126 (30.4)
	T3	26 (7.1)	12 (26.1)	38 (9.2)
	T4	3 (0.8)	3 (0.7)	6 (1.5)
Final N-stage				
	pN0	282 (76.6)	-	282 (68.1)
	pN1	86 (23.4)	-	86 (20.8)
	pN2	-	28 (60.9)	28 (6.8)
	pN3	-	18 (3.9)	18 (4.4)

**Table 2 cancers-15-01859-t002:** Cross-tabulation of MR findings and N-stage.

	Significant Nodal Disease Absent		Significant Nodal Disease Present	Total
	pN0	pN1	pN2 or Higher	
MRI positive	3	38	42	83
MRI negative	279	48	4	331
Total	282	86	46	414

**Table 3 cancers-15-01859-t003:** Diagnostic indices. TP = true positives, FP = false positives, FN = false negatives, TN = true negatives, PPV = positive predictive value, NPV = negative predictive value, No = number of women.

Test Accuracy Measurements	Diagnosis of Any Nodal Disease (Final Stage >pN0)	Diagnosis of Significant Nodal Disease (Final Stage ≥N2)
TP (No.)	80	42
FP (No.)	3	41
FN (No.)	52	4
TN (No.)	279	327
Sensitivity	80/132 (60.6%; 95% CI: 51.7–69.0)	42/46 (91.3%; 95% CI: 79.2–97.6)
Specificity	279/282 (98.9%; 95% CI: 96.9–99.8)	327/368 (88.7%; 95% CI: 85.2–91.9)
NPV (%)	279/331 (84.2%; 95% CI: 81.3–86.9)	327/331 (98.8%; 95% CI: 97.0–99.5)
PPV (%)	80/83 (96.4%; 95% CI: 89.6–98.8)	42/83 (50.6%; 95% CI: 43.1–58.1)

## Data Availability

The data presented in this study are available on request from the corresponding author. The data are not publicly available due to ethical restrictions.

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
