# Peer review of "Abbreviated MRI for Comprehensive Regional Lymph Node Staging during Pre-Operative Breast MRI"

_cancers, 2023, doi:10.3390/cancers15061859_

Round 1

Reviewer 1 Report

Aim of this study was to investigate whether such a short pulse sequence is useful to 84  help identify patients without significant LNM who can proceed according to the ACO- 85 SOG Z0011 pathway. This work is of great clinical significance for the detection of comprehensive regional lymph node 2 staging. A few minor concerns should be addressed appropriately before consideration for publication.

1.      What are the difficulties and thorny problems of MRI in the detection of comprehensive regional lymph node 2 staging?

2.      The author should point out the significance and value of this work for clinical research.

3.      The clarity of the article pictures is not clear.

4.      Some imaging related papers or reviews should be cited (e.g., Journal of Bio-X Research, 2020, 3(04): 144-156; Wiley Interdisciplinary Reviews: Nanomedicine and Nanobiotechnology 14.4 (2022): e1797; Nano letters, 2019, 19(11): 8234-8244.).

Author Response

Response to Reviewer 1

Reviewer #1: “Aim of this study was to investigate whether such a short pulse sequence is useful to help identify patients without significant LNM who can proceed according to the ACO- 85 SOG Z0011 pathway. This work is of great clinical significance for the detection of comprehensive regional lymph node 2 staging.”

Authors’ Response: We would like to thank the Reviewer for taking the time to review our manuscript and the equally insightful and constructive comments. Please see our responses to the specific comments below.

R1 Specific Comment #1: What are the difficulties and thorny problems of MRI in the detection of comprehensive regional lymph node 2 staging?

In our study, MRI was false-negative in 4/46 women with significant nodal disease (≥pN2). Even in the knowledge of substantial lymph node metastases (up to 10 lymph node metastases) in the axilla, a review of the false-negative cases revealed no suspicous morphology or enlargement of the lymph nodes. A possible explanation is, that in the 8th edition of TNM classification of malignant tumours cancer cells in four to nine lymph nodes in the axilla, and at least one is larger than 2 mm are classified as pN2. Therefore, overall tumor burden within the metastatic lymph nodes can be relatively small and not be detectalbe in MR imaging.

We have now added this detail of the TNM classification to the material and methods section (line 187-188) and discussed this aspect (line 333-336).

R1 Specific Comment #2: The author should point out the significance and value of this work for clinical research.

Done as suggested (line 337-343).

R1 Specific Comment #3: The clarity of the article pictures is not clear.

We have now uploaded high-resolution figures.

R1 Specific Comment #4: Some imaging related papers or reviews should be cited (e.g., Journal of Bio-X Research, 2020, 3(04): 144-156; Wiley Interdisciplinary Reviews: Nanomedicine and Nanobiotechnology 14.4 (2022): e1797; Nano letters, 2019, 19(11): 8234-8244.).

Thanks for pointing out these interesting articles. However, we did not believe that these articles support the message of our research work. In the current reference list, reviews and imaging related papers are represented. No changes were done.

Reviewer 2 Report

Dear authors,

Thank you for submitting your original research manuscript cancers-2205033, under consideration for publication in Cancers, a journal of MDPI (IF 6.73/Q1 Oncology).

The manuscript is a retrospective cross-sectional diagnostic accuracy study from the field of breast MRI in breast cancer staging. The authors test the hypothesis that the MR-based lymph node staging could predict relevant (>= N2) lymph node metastasis in subjects with breast cancer before the operation. For this purpose, the authors appended the standard breast-MRI protocol (1,5T Philips Achieva) by a 3-min coronal T1w non-contrasted, non-fat-sat sequence over the chest lymph node levels (axillar, parasternal, supraclavicular) using the built-in gantry coil as a receiver. The concept is to perspectively engage MRI as the standard-of-care for dichotomizing patient selection for radical lymphadenectomy.

The topic is clinically attractive and relevant to the aims and scopes of Cancers. Methods are appropriately described, and the whole manuscript is compiled in concise, professional language.

It is strongly recommended that the authors revise their text according to the following comments:

C001 – the standard of care for LN-staging preoperatively is the ultrasound of lumph node basins, thich is on the one hand user-dependend, as the authors explain, but also more available and cost-effective for the healthcare system. The reviewer believes that the authors should insist more in the comparison of their finding and cost burdens with the numerous ultrasound-based studies, especially concerning the weakest point of their study: the false positives.

C002 – MRI-based staging has as a significant drawback the high prevalence of false positives (PPV ca. 50% - equal to tossing a coin). However, the high NPV allows for suggesting MRI as a “rule out” method to select patients for a further workout.

In times of financial challenge in the healthcare system, it is imperative for the persuasive potential and the citability of this study that the “rule out” potential of such an expensive method such as the MRI is superior to the current standard-of-care, the ultrasound.

C003-Line116: “prospectively”? or retrospectively? Please revise.

C004-L128: “similar signal intensity” – where do you set your threshold for defining signal intensity differences of LN compared to contralateral?

C005 – Table1>Women with significant LN>

-          Premenopausal and postmenopausal> percentages do not add up to 100%. Please correct or explain.    

-          T-stage> percentages do not add up to 100%. Please correct or explain

-          Final N-stage> percentages do not add up to 100%. Please correct or explain

Author Response

Response to Reviewer 2

Reviewer #2: “The manuscript is a retrospective cross-sectional diagnostic accuracy study from the field of breast MRI in breast cancer staging. The authors test the hypothesis that the MR-based lymph node staging could predict relevant (>= N2) lymph node metastasis in subjects with breast cancer before the operation. For this purpose, the authors appended the standard breast-MRI protocol (1,5T Philips Achieva) by a 3-min coronal T1w non-contrasted, non-fat-sat sequence over the chest lymph node levels (axillar, parasternal, supraclavicular) using the built-in gantry coil as a receiver. The concept is to perspectively engage MRI as the standard-of-care for dichotomizing patient selection for radical lymphadenectomy.

The topic is clinically attractive and relevant to the aims and scopes of Cancers. Methods are appropriately described, and the whole manuscript is compiled in concise, professional language.”

Authors’ Response: We would like to thank the Reviewer for taking the time to review our manuscript and the equally insightful and constructive comments. Please see our responses to the specific comments below.

R2 Specific Comment #1: The standard of care for LN-staging preoperatively is the ultrasound of lymph node basins, which is on the one hand user-dependent, as the authors explain, but also more available and cost-effective for the healthcare system. The reviewer believes that the authors should insist more in the comparison of their finding and cost burdens with the numerous ultrasound-based studies, especially concerning the weakest point of their study: the false positives.

We agree with the reviewer that cost efficiency is an important aspect in healthcare – which is the exact reason why we do research on abbreviated MRI. And it is certainly true that ultrasound is more available than MRI. However, ultrasound of the axilla is an additional examination for patients who undergo breast MRI routinely for pre-operative treatment planning. Based on our results, a very short additional pulse sequence – which leads to a negligible prolongation of the regular breast MRI protocol – allows accurate lymph node staging – and not only of the lymph node basins of the axilla (which are amenable to ultrasound), but also the parasternal region and the supra- and infraclavicular basins. So, without additional costs, we provide information that can replace ultrasound, and we provide information that axillary ultrasound cannot provide. To the best our our knowledge, there is no evidence on the cost effectiveness of axillary ultrasound.

We clarified throughout the summary and manuscript that this short additional pulse sequence was part of a pre-operative routine breast MRI and not an additional examination. Please see the manuscript for these changes.

R2 Specific Comment #2: – MRI-based staging has as a significant drawback the high prevalence of false positives (PPV ca. 50% - equal to tossing a coin). However, the high NPV allows for suggesting MRI as a “rule out” method to select patients for a further workout.

In times of financial challenge in the healthcare system, it is imperative for the persuasive potential and the citability of this study that the “rule out” potential of such an expensive method such as the MRI is superior to the current standard-of-care, the ultrasound.

We would like to draw the attention of the reviewer to the fact that a PPV (positive predictive value of biopsy recommendations) of 50% is exceptionally high. Since the reviewer is proposing axillary ultrasound: Ultrasound is in fact the imaging method that is associated with the lowest PPVs. The PPV of breast ultrasound is between 8% and 12% - which is far worse than tossing a coin; as a matter of fact, it means that 92% of the biopsies made because of suspicious ultrasound findings are indeed benign. No changes were done.

R2 Specific Comment #3: “prospectively”? or retrospectively? Please revise.

Thanks for pointing this out. We corrected the error accordingly (line 120).

R2 Specific Comment #4: “similar signal intensity” – where do you set your threshold for defining signal intensity differences of LN compared to contralateral?

The signal intensity was visually compared to the respective contralateral lymph nodes. We have clarified this in the material and methods section (line 129). More detailed information can be found in Material and Methods – MRI Evaluation of Lymph Node Status (line 130-145)

R2 Specific Comment #5

– Table1>Women with significant LN
- Premenopausal and postmenopausal> percentages do not add up to 100%. Please correct
 or explain.
- T-stage> percentages do not add up to 100%. Please correct or explain
- Final N-stage> percentages do not add up to 100%. Please correct or explain

Thank you for pointing this out. The errors have been corrected accordingly. For changes please refer to Table 1.

Reviewer 3 Report

Thank you very much for inviting me to review this very interesting paper.

I have only a few comments:

Abstract: line 23 - should be "women"

Introduction: line 43. I would suggest not to start introduction with "Although", and also suggest to shorten this first sentence that is too long.

Line 139 - negative (the e is missing)

Results: line 200: preoperative - the e is missing

Thank you very much

Author Response

Response to Reviewer 3

Reviewer #3: “Thank you very much for inviting me to review this very interesting paper.”

Authors’ response: We would like to thank the Reviewer for taking the time to review our manuscript.

Please see our responses to the specific comments below.

R3 Specific Comment #1: Abstract: line 23 - should be "women"

Thank you for pointing this out. The typo has been corrected accordingly (line 28).

R3 Specific Comment #2: Introduction: line 43. I would suggest not to start introduction with "Although", and also suggest to shorten this first sentence that is too long.

Agreed. The first sentence has been reformulated and shortened (lines 48-52).

R3 Specific Comment #3: Line 139 - negative (the e is missing)

Thank you for pointing this out. The typo has been corrected accordingly (line 155). 

R3 Specific Comment #4: Results: line 200: preoperative - the e is missing

Thank you for pointing this out. The typo has been corrected accordingly (line 204). 

Reviewer 4 Report

Introduction

OK

M&M

MR imaging protocol: please provide the scan time for the coronal T1 sequence 

Results

OK

Discussion 

OK

Author Response

Response to Reviewer 4

Author: We would like to thank the Reviewer for taking the time to review our manuscript and the overall appreciation.

R4 Specific Comment #1: MR imaging protocol: please provide the scan time for the coronal T1 sequence.

Done as suggested (line 112-113).

Round 2

Reviewer 1 Report

The author made very detailed modifications according to my opinions. Therefore, I recommend that this manuscript be published in its current form.